# Learning Relational Invariance for Out-of-Distribution Molecular Relational Learning

## Abstract

Molecular Relational Learning (MRL) expands the scope of molecular representation learning by incorporating additional molecules, aiming to understand the interactions between pairs of molecules. While MRL has shown promising results, the existing methods have not been able to generalise to real world scenarios. Invariant learning is pivotal in addressing Out-of-Distribution (OOD) generalization challenges. However, two major obstacles impede the progress of invariant learning in MRL: (1) Unlike single-molecular cases, interactions between molecules introduce added complexity, with a heavy reliance on molecular substructure recognition, often leading to the misspecification of invariant patterns. (2) Accurate modeling of interactions can effectively improve generalizations. However, previous methods focus on node interactions, which is limited by the expressiveness of GNN, and long-range interactions cannot be captured. To address these, we propose a novel Relational Invariant Learning (RIL) framework that uses a multi-granularity interaction approach to improve OOD generalization for MRL, and the framework is denoted as RILOOD. Specifically, we model the environment diversity distribution of molecules by mixup-based Conditional Modeling. Then, we employ a multi-granularity refinement strategy to learn the Context-Aware Representation, which is essential for capturing multi-level interaction. We further design an invariant learning module to capture the invariant patterns that robustly generalize across unseen environments. Extensive experiments on molecular datasets show that our method achieves stronger generalization against state-of-the-art methods in the presence of various distribution shifts. Our code will be released after our paper is accepted.

## 1 Introduction

Predicting molecular properties in solvent is crucial, given that most chemical and biological processes occur in solution. Solvent-based molecular property prediction, also referred to as Solute-Solvent Interaction in Molecule Relational Learning (MRL)(Lim & Jung, 2019; Subramanian et al., 2020; Panwar et al., 2021; Low et al., 2022; Zhang et al., 2022; Lee et al., 2023a;b), has played a pivotal role in chemical and biological research, including battery manufacture, pharmaceutical industry (Chung et al., 2022; Varghese & Mushrif, 2019). It is an evolving field that aims to understand interactions between solutes and solvents at the molecular-level, allowing for predicting molecular property through a prior. More importantly, it significantly extends the conventional molecular property prediction practices by taking solvent molecular as additional inputs, thereby achieving promising performance and chemical interpretability.

Despite their notable success, existing methods are based on the assumption that training and test data are sampled from an independent and identical distribution (I.I.D.). However, the real world is open, diverse, and uncertain. Out-of-Distribution (OOD) refers to scenarios where the test data or new data encountered by a model significantly differ from the training data. For single molecule, OOD can occur not only in the molecule structure itself—such as differences in size or scaffold—but also in the target properties. OOD generalization(Krueger et al., 2021), which seeks to address this challenge by learning invariant representations across multiple environments (e.g., scaffolds, sizes), has garnered significant attention. Typically, the privileged substructure remains invariant concerning a molecular's properties. However, one important nature of solvated molecules is the non-stationary property, indicating that its statistical features are changing over solvent. As shown in Fig.1, previous

Figure 1: A toy example shows the 'solute-solvent interactions' with distribution shifts when the underlying environments change (e.g., solvent). A model could mistakenly predict that strong polar molecules are easily soluble in polar solvents and not true for low polar molecules if it fails to capture interaction invariant patterns among spurious correlations.

methods would spuriously correlate non-causal factors ('substructure') and produce undesired results under a new environment. Scaffolds and size, etc., are often considered to be irrelevant patterns to molecular properties, which can be seen as spurious correlations.

Existing works mainly attempt to build effective methods for distribution shifts, from invariant learning(Wu et al., 2022a), feature disentanglement(Liu et al., 2021), to data augmentation(Sui et al., 2024; Jia et al., 2024). Thus far, few previous works focus on OOD generalization on MRL. A typical work(Lee et al., 2023b) is devised to solve the distribution shift problem relies on the identification of molecule substructure by causal inference. Nevertheless, the complicated molecular pairs interaction, which are largely underexplored in graph invariant representations, makes it challenging to accurately distinguish the invariant causal parts from diverse spurious correlations. On the other hand, mis-specification refers to variant or spurious correlations that cannot be invariant of the any available environments(e.g., a toy example in Fig. 1). It is hoped that a new approach will be developed to facilitate the generalization of molecular properties toward open-scenario.

To address these limitations, in this work, we propose a novel Relational Invariant Learning framework against Out-of-Distribution Generalization in MRL. In contrast to the traditional methods, we present a novelty framework to capture the invariance in molecular pairs and achieve generalized representation. Specifically, we first employ GNN to encode molecular, following by the cross-attention module to map atom-level interaction. Then, we utilize mixup-enhanced Conditional Variational Modeling. We embrace the strengths of cross-environment invariance by considering a multi-granularity context-aware interaction and environment diversity inference. Learn interaction invariance(Xie et al., 2024), which helps to uncover the underlying relationships between molecules in a chemically interpretable way in latent space.

Our main contributions can be summarized as follows: (1) We propose a novel Relational Invariant Learning framework, call RILOOD, to solve the OOD generalization on molecular relational learning. (2) Our method not only preserves the fine-grained interactions between molecules at the molecular-level, but also captures the global interaction information through multi-granularity context-aware refinement. (3) We formulate the OOD generalization problem on MRL. Focusing on both invariant interaction learning and conditional modeling, capturing associations between different distributions through domain shift. It exhibit robustness and transferability across different data domains.

## 2 RELATED WORKS

### 2.1 MOLECULAR RELATIONAL LEARNING

Molecular Relational Learning(Lim & Jung, 2019; Subramanian et al., 2020; Lee et al., 2023a;b; Pathak et al., 2020), which aims to study the relationship between moleculars, can be divided into molecular interaction prediction and Drug-Drug Interaction prediction. Molecular interaction

prediction, i.e. solvent-based molecular property prediction, includes solvent free energy prediction, solubility prediction, chromophore absorption prediction, and so on. Unlike traditional molecular property prediction, the model need predict the properties exhibited by the same molecular exposed to multisolvent. Recent works leverage message-passing network to encode atomic representations and further improving the interpretability of model using an interaction map(Lee et al., 2023a;b; Pathak et al., 2020).

## 2.2 OUT OF DISTRIBUTION GENERALIZATION

Generalizing well-trained method to unseen environment with different data distributions is challenging and promising problem on machine learning due to wide applicability. Current state-of-the-art approaches can be roughly categorized into three types. (1) Invariant learning method. There are plentiful studies in invariant learning without environment labels. However, ZIN(Lin et al., 2022) argue that it is impossible to identify the invariant features without given environment labels in Euclidean data, and propose to leverage additional auxiliary information for invariant learning. (2) Causal inference theories utilize Structural Causal Model (SCM)(Chen et al., 2022; Lu et al., 2021) or Independent Causal Mechanism (ICM)(Peters et al., 2017; Gui et al., 2024) assumption to filter out spurious correlation and strengthen the invariant causal patterns. (3) Disentangled learning requires strong prior assumptions that can effectively separating semantic factors into two categories: (i) invariant features that consistently predict the label across distributions, and (ii) spurious features that have unstable correlations with the label. Current methods are mainly to discover and define the invariant factors in the data collection process, and design effective algorithms based on the invariance to guide the model to achieve out-of-distribution generalization.

## 2.3 INVARIANT LEARNING IN MOLECULAR RELATIONAL LEARNING

Current research on invariant learning in MRL prediction is still sparse. Among these works, one scheme is the identification of the core substructure(Lee et al., 2023a), which involves utilizing the minimum sufficient information related to the task according to the principle of graph information bottleneck. Another scheme(Lee et al., 2023b) proposes to learn causal substructure using causal intervention to solve distribution shift. In OOD scenario, assessing model generalization typically involved dividing datasets into scenarios like "unseen solvent" or "unseen domain", where the test set exclusively contained certain bias. However, previous evaluations often remain within the intra-domain framework, which does not fully align with real-world conditions. Despite invariant learning success on graph(Wu et al., 2022a; Yang et al., 2022; Li et al., 2022), it still be confined on graph pairs by two critical limitations: (1) Different from Euclidean data such as image, the environmental label of the graph is not easy to obtain. The existing environment is handcrafted or rule-based, not structured, which could provide insufficient information for capturing the fundamental relations across domains from the casual data-generating perspective. (2) Invariant patterns, spurious correlations are entangled with shortcuts, and latent invariant representations are not easy to decouple.

## 3 PRELIMINARIES

We define the uppercase letters (e.g., $\mathcal{G}$) as random variables, the lower-case letters (e.g., $g$) are samples of variables, and the blackboard typefaces (e.g., $\mathbb{G}$) denote the sample spaces. Let $\mathcal{G} = (\mathcal{V}, \mathcal{E}) \in \mathbb{G}$ denotes as a graph, where $\mathcal{V} = \{v_1, v_2, ..., v_n\}$ is the set of nodes and $\mathcal{E} \in \mathcal{V} \times \mathcal{V}$ is the set of edges.

### 3.1 NOTATIONS AND PROBLEM FORMULATION.

The goal of MRL task is to predict the target label Y given the associated input molecule pairs $(\mathcal{G}_1, \mathcal{G}_2)$. It can be formulated as modeling the conditional distribution $p(\mathcal{G}_1|\mathcal{G}_2)$.

**Problem formulation.** Given a dataset $\mathcal{D} = \{((\mathcal{G}_1^i, \mathcal{G}_2^i), Y^i)\}_{i=1}^N$, where $\mathcal{G}_1$ is solute molecule, and $\mathcal{G}_2$ is solvent molecule, each molecule pairs is associated with a target label $Y$. N is the total number of dataset. The objective is to train a model to predict Y based on the input $(\mathcal{G}_1, \mathcal{G}_2)$. The model should effectively learn the relationships between the input features and the target variable, leveraging the information from both $\mathcal{G}_1$ and $\mathcal{G}_2$ to accurately predict Y. The model's performance will be evaluated based on the RMSE of the predicted output $\hat{Y}$ in comparison to the true labels Y.

**Molecular Representation.** We implement our method based on Pathak et al. (2020), which is a message passing architecture devised for the solute and solvent molecule interaction. Given a pair of molecules $\mathcal{G}_1 = (\mathcal{V}_1, \mathcal{E}_1)$ and $\mathcal{G}_2 = (\mathcal{V}_2, \mathcal{E}_2)$. We first obtain the node representation of each molecular as follows: $h_1 = \text{GCN}(\mathcal{V}_1, \mathcal{E}_1), h_2 = \text{GCN}(\mathcal{V}_2, \mathcal{E}_2)$. To capture the inter-molecular interaction in node-level, the interaction map is constructed as following: $I = h_1 \cdot h_2^T$, where $\cdot$ is matrix multiplication, $I \in \mathbb{R}^{N_1 \times N_2}$. We obtained a representation $\tilde{h}_1 \in \mathbb{R}^{N_1 \times D}$ of the solvent's interaction on the solute and a representation $\tilde{h}_2 \in \mathbb{R}^{N_2 \times D}$ of the solute's interaction on the solvent through a shared interaction map according to the following equations: $\tilde{h}_1 = I \cdot h_2, \tilde{h}_2 = I^T \cdot h_1$. Here, $N_1$ and $N_2$ denote the number of atoms in molecule $G_1$ and $G_2$, respectively. $h_1$ is generated by concatenating two representation $\tilde{h}_1$ and $h_1$, i.e. $H_1 = concat[h_1, \tilde{h}_1]$. The overall graph representation is obtained using a readout layer $R_{solute}(H_1)$, which set the READOUT function as Set2Set(Vinyals et al., 2015).

### 3.2 OOD GENERALIZATION.

In this work, we mainly focus on OOD generalization in graph-level prediction tasks. Our aim is to train the model with limited label to infer the domain distribution from unseen data in $\mathcal{D}_{te}$.

**Problem formulation.** Given a molecular pairs datasets, $\mathcal{D} = \{((G_1^i, G_2^i), Y^i)_{i=1}^{N^{tr+te}}\}$ collect from multiple environments $\mathcal{E}$, which were considered as drawn independently from an identical distribution $P_e$, i.e., $\mathcal{D}_{ID} = \{(G_1, G_2) \in \mathcal{D} \mid G_1 \in G_{ID} \bigwedge G_2 \in G_{ID}\}$. The training and test datasets are denoted as $\mathcal{D}_{tr} = \{((G_1^i, G_2^i), Y^i)\}_{i=1}^{N^{tr}}$ and $\mathcal{D}_{te} = \{((G_1^i, G_2^i), Y^i)\}_{i=1}^{N^{te}}$. Our goal is to find an optimal predictor $\Phi: (\mathbb{G}_1, \mathbb{G}_2) \to \mathbb{Y}$ that performs well on all environments. Formally, the learning objectives can be formulated as:

$$\min_f \max_{e \in \mathcal{E}} \mathbb{E}_{((G_1^i, G_2^i), Y^i) \sim p((\mathbf{G_1}, \mathbf{G_2}), \mathbf{Y} \mid N=e)} \left[ \ell \left( \Phi \left( G_1^i, G_2^i \right), Y^i \right) \mid e \right] \tag{1}$$

**Definition 1.** *(Data generation process) The OOD distribution can be sampled according to $\mathcal{D}_{OOD} = \{(G_1, G_2) \in \mathcal{D} \mid (G_1 \in G_{OOD} \bigwedge G_2 \in G_{OOD}) \bigvee (G_1 \in G_{OOD} \bigwedge G_2 \in G_{ID}) \bigvee (G_1 \in G_{ID} \bigwedge G_2 \in G_{OOD})\}$. The data generation process is as follows: Let $\mathcal{E}$ denote all possible environments, $supp(N_{tr}) \subset supp(\mathcal{E})$, sampled train data from $P((G_1, G_2), Y)$. Distribution shifts indicate that $P_e((G_1, G_2), Y) \neq P'_e((G_1, G_2), Y)$, i.e., $\mathcal{D}_{Train} = \{((G_1^i, G_2^i), Y^i)_{i=1}^{N^{tr}} \mid e \subset supp(N_{tr})\}$, $\mathcal{D}_{Test} = \{((G_1^i, G_2^i), Y^i)_{i=1}^{N^{te}} \mid e' \in supp(\mathcal{E}) \backslash supp(N_{tr})\}$.*

## 4 METHODOLOGY

In this section, we present the details of **RILOOD**, an **R**elational **I**nvariant **L**earning framework, to solve the **O**ut-**o**f-**D**istribution generalization on molecular relational learning. An overview of the proposed method is shown in Fig. 2. We illustrate three key components in RILOOD, i.e., Mixup-enhanced Conditional Variational Modeling, Multi-granularity Context-Aware Refinement, Invariant Ralational Learning Mechanism.

### 4.1 INVARIANT LEARNING ON RELATIONAL LEARNING.

The goal of the invariant-based approach is to train a predictor that is robust to distribution changes, i.e., a mapping from molecular pairs to label that does not vary with environment. It is hoped that the predictor will be able to satisfies the following two properties:

**Assumption 1.** *Given the molecular pairs $(\mathbb{G}_1, \mathbb{G}_2)$, each molecular pairs is associated with $K$ surrounding environments. There exist invariant interaction patterns that can lead to generalized out-of-distribution prediction across all environment slices. The optimal representation learner $\Phi(\cdot)$ satisfying:*

*(1) Invariance Property:$\forall e, e' \in supp(\varepsilon)$, $P(Y^e \mid H^e, e) = P(Y^{e'} \mid H^{e'}, e')$, where $H^i = \Phi(\mathcal{G}_1^i, \mathcal{G}_2^i)$ denotes molecular pairs representations, $H^e = \Phi(\mathcal{G}_1^e, \mathcal{G}_2^e)$, $H^{e'} = \Phi(\mathcal{G}_1^{e'}, \mathcal{G}_2^{e'})$;*

*(2) Sufficiency Property:$Y^i = f(\Phi(\mathcal{G}_1^e, \mathcal{G}_2^e)) + \epsilon$, where $f$ is a predictor, $\epsilon$ is a random noise.*

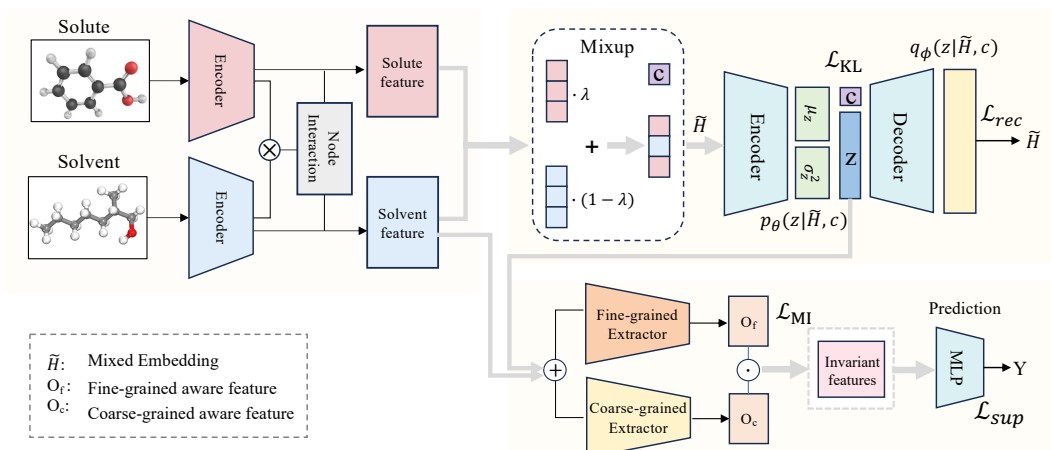

Figure 2: Overall the framework of RILOOD.

*If the following conditions hold: (1) $\Phi(\mathcal{G}_1, \mathcal{G}_2) \perp (\mathcal{G}_1, \mathcal{G}_2) \backslash \Phi(\mathcal{G}_1, \mathcal{G}_2)$; (2) $\forall \Phi \in supp(\mathcal{E}), \exists e' \in supp(\mathcal{E})$ such that $P^{e'}(\mathcal{G}_1, \mathcal{G}_2, Y) = P^{e'}(\Phi(\mathcal{G}_1, \mathcal{G}_2), Y) P^{e'}((\mathcal{G}_1, \mathcal{G}_2) \backslash \Phi(\mathcal{G}_1, \mathcal{G}_2))$ and $P^{e'}(\Phi(\mathcal{G}_1, \mathcal{G}_2)) = P^e(\Phi(\mathcal{G}_1, \mathcal{G}_2))$.*

Specifically, We further decompose $\Phi(\cdot) = g \odot h(\mathcal{G}_1, \mathcal{G}_2)$ by two sub-components: (a) a Conditional Variational AutoEncoder (CVAE) $h$: infer the distribution of solute $H_1 \sim q_\phi(H_1|z, c)$ across environment $c$; (b) a Multi-granularity Context-Aware Learner $g : (H_1, H_2) \rightarrow H_{12}$ aiming to identify the desired $H_{12}$. Based on Eq. 1, we can reformulate the OOD problem on molecular pairs as:

$$\min_f \max_{\mathbf{e} \in \mathcal{E}} \mathbb{E}_{(\mathcal{G}_1^i, \mathcal{G}_2^i, Y^i) \sim p(\mathcal{G}_1, \mathcal{G}_2, \mathbf{Y}|e)} \left[ \ell \left( g \odot h \left( \mathcal{G}_1^i, \mathcal{G}_2^i \right), Y^i \right) \mid e \right], \quad (2)$$

where $e$ denotes the support environments, $\Phi(\cdot)$ is the representation learner and $\ell(\cdot, \cdot)$ represents a loss function.

## 4.2 MIXUP-ENHANCED CONDITIONAL VARIATIONAL MODELING

The ability to generalise to unseen distributions is guaranteed by a predictor that performs well in several predefined environments. Theoretically, spurious patterns can be used to infer the underlying environment. Lin et al. (2022) proposed that environment partitioning can be learned using additional auxiliary information to separate invariant features. Indeed, we fail to obtain environment labels of molecular pairs directly. Consequently, we utilize auxiliary information as a condition, such as solvent, to model the distribution of molecules across domains.

The Conditional Variational AutoEncoder(CVAE) has been widely adopted for modeling conditional distributions in latent environments through multi-label variational inference. We propose Mixup-based CVAE (MCVAE) specifically designed to model molecular distribution using the paired solvent information and infer $q_\phi(z|\mathcal{G}_1, c)$ across various environments. At the same time, the uncertainty constraint is added. Assume that the categories of solvent are K, i.e., $C = \{c^k\}_{k=1}^K$. Each type of solvent $c^k$ is represented as a K-dimensional one-hot column vector $c^k \in \{0, 1\}^K$ whose k-th dimension is 1. Mixup techniques generate a variety of environment data to help models generalize to unseen domains. We obtain the molecular representations $H_1$ and $H_2$ for molecule $\mathcal{G}_1$ and molecule $\mathcal{G}_2$ in the previous part. Next, we apply mixup to the obtain augmentation sample as follow:

$$\tilde{H} = \lambda \cdot H_1 + (1 - \lambda) \cdot H_2, c = \lambda \cdot c_1 + (1 - \lambda) \cdot c_2 \quad (3)$$

where $\tilde{H}$ is the mixed representation of $H_1$ and $H_2$. c is the mixed label of $c_1$ and $c_2$. $\lambda \in [0, 1]$ is drawn from a Beta distribution, i.e., $\lambda \sim Beta(\alpha, \alpha)$. Specifically, we introduce variational inference to generate loglikelihood $logp(\tilde{H} \mid c)$, which can be reformulated as the following variational lower bound by introducing the approximate posterior distribution $q(z, c)$.

$$\max_{\theta, \phi} \mathbb{E}_{\tilde{H} \sim D} \left[ \mathbb{E}_{q_\phi(z|\tilde{H}, c)} \left[ \log p_\theta(\tilde{H} \mid z, c) \right] \right], \text{s.t.} D_{KL} \left( q_\phi(z \mid \tilde{H}, c) || p_\theta(z \mid \tilde{H}) \right) < \epsilon \quad (4)$$

By optimizing MCVAE, we aim to infer the molecular distribution of the latent environment, a novel approach to learning environment-sensitive molecular representations. We minimize the difference between the approximate distribution $q(z|\tilde{\mathrm{H}}, c)$ of the latent variable z and the true posterior probability $p(\tilde{\mathrm{H}}|c, z)$ for a specific solvent c. Leveraging rich prior knowledge from conditional encoders, the training objectives are, (1) make the K-group normal distribution output by the encoder as close as possible to the standard normal distribution; (2) We adopt the Monte Carlo method by drawing samples $z^{(\ell)}(\ell = 1, 2, ..., \mathcal{L})$ from the distribution $q(z|c)$, which make the resampled solute molecular features as close as possible to the original features. Maximizing the conditional log-likelihood $logp_\phi(z|c)$ leads to an optimal MCVAE by minimizing:

$$\mathcal{L}_{\text{MCVAE}}(\theta, \phi; \tilde{H}, \mathrm{c}) = -KL\left(q_\phi(\mathrm{z}^{(l)} \mid \tilde{H}, \mathrm{c})\|p_\theta(\mathrm{z} \mid \tilde{H})\right) + \frac{1}{N}\sum_{i=1}^{N}\left[\frac{1}{\sigma(\tilde{H})^2}\|z - \tilde{H}\|^2 + \log\sigma(\tilde{H})^2\right]$$

(5)

where $\mathrm{z}^{(l)} = g_\phi\left(\tilde{H}, \mathrm{c}, \epsilon^{(l)}\right), \epsilon^{(l)} \sim \mathcal{N}(0, \mathrm{I})$and L is the number of samples. Here, for the regression task, we introduce an uncertainty constraint to reduce the additional noise introduced by the mixing technique. Detailed proofs are in Appendix A.1.

### 4.3 MULTI-GRANULARITY CONTEXT-AWARE REFINEMENT.

In preliminary 3.1, molecular interactions are constructed using cross-attention on node-level features. However, existing methods are limited by the expressiveness of vanilla GNNs, which tend to be over-smoothed. Therefore, it is not easy to distinguish subtle differences. Additionally, existing approaches are grounded in molecular invariant learning, which relies heavily on the core substructures of the molecule, leading to inherent biases. Motivated by the fact that there are non-chemically bonded interactions between molecules, we employ self-attention mechanism to identify invariant features.

Consequently, the interactions are modeled utilizing sampled features $\mathrm{z}^{(l)}$ and solvent features $H_2$, and further propose a Multi-granularity Context-Aware Refinement (MCAR) strategy to capture multi-level interactions at the graph level, including: (1) fine-grained context interactions across each dimension, and (2) coarse-grained context interactions for each molecular graph. Specifically, let $\mathrm{z}^{(l)}$, $H_2$ denote as solute molecule embedding and solvent molecule embedding, respectively.

$$\mathrm{Q}, \mathrm{K}, \mathrm{V} = EW^Q, EW^K, EW^V \tag{6}$$

where $\mathrm{E} = concat[\mathrm{z}^{(l)}, H_2]$, $W^Q, W^K, W^V \in \mathbb{R}^{d_k}$ are transformation matrices, and $d_k$ is the attention size. We develop the MCAR mechanism by two steps: (1) Capture coarse-grained molecular-level contexts and fine-grained feature-level contexts to learn context information together; (2) The invariant patterns are updated by matrix multiplication between coarse-grained features and fine-grained features.

$$\mathrm{O}_c = Attention(\mathrm{Q}, \mathrm{K}, \mathrm{V}) = \text{Softmax}\left(P_\ell\frac{QK^T}{\sqrt{d_k}}\right)VP_w \in \mathbb{R}^{f\times d}$$

$$\mathrm{O}_f = PReLU\left(\mathrm{W}_L\mathrm{h}_l + b_l\right) \in \mathrm{R}^{1\times d} \tag{7}$$

$$\mathrm{H}_c = \mathrm{O}_c \circ \mathrm{O}_f \in \mathbb{R}^{f\times d}$$

where Q, K, V are given by Eq. 6, and $P_\ell \in \mathbb{R}^{d_k\times d_k}, P_W \in \mathbb{R}^{d_k\times d_v}$ are the two additional linear projections. Self-attention is suitable for extracting relationships between molecules, while fine-grained interactions can be used to extract contextual information from different instances using a simple linear layer. Each layer of the MLP is obtained as follows: $h_{l+1} = PReLU(W_lh_l + b_l)$. To enhance effective feature extraction, maximizing mutual information allows for the retention of important features while minimizing redundancy and noise. We maximize the mutual information between $E$ and $\widehat{H}_{inv}$. $E$ is the global feature of merging solute and solvent and is dominated by spurious correlations, while $\widehat{H}_{inv}$ is the context-aware feature dominated by invariant correlations.

$$\max_{f_c, w} I\left(\widehat{H}_{inv}; Y\right), \text{s.t.}\widehat{H}_{inv} \in \argmax_{\widehat{H}_{inv}=w(E), \left|\widehat{H}_{inv}\right|\leq E} I\left(\widehat{H}_{inv}; E \mid Y\right) \tag{8}$$

Finally, contrastive learning provides a practical solution for the approximation, the learning objective is defined as

$$\mathcal{L}_{MI} = -\frac{1}{M}\sum_{i=1}^{M}\log\frac{exp(sim(\hat{H}_{inv}, E^i))}{exp(sim(\hat{H}_{inv}, E^i)) + \sum_{j=1, j\neq i}^{M} exp(sim(\hat{H}_{inv}, E^j))} \quad (9)$$

### 4.4 INVARIANT RELATIONAL LEARNING MECHANISM

**Optimization Objective.** Eq. 2 clarifies the training objective of OOD generalization. However, directly optimizing Eq. 2 is not impracticable. Specifically, we jointly optimize objectives:

$$\mathcal{L} = \mathcal{L}_{\text{inv}} + \alpha\mathcal{L}_{\text{MCVAE}} + \beta\mathcal{L}_{\text{MI}} \quad (10)$$

where $\alpha$ and $\beta$ are weight hyperparameters for $\mathcal{L}_{\text{MCVAE}}$ and $\mathcal{L}_{\text{MI}}$ , respectively. The $\mathcal{L}_{\text{inv}}$ calculates the loss between the model prediction given the pair of input graphs, i.e., $(G_1, G_2)$, and the target.

**Proposition 1.** *Given observed environment label $c$, our goal is to build a model $p_\theta(\tilde{H}|c, z)$ that learns the feature $\tilde{H} \in \mathbb{R}^{N_x}$ conditioned on $c$. Optimizing Eq. 12 letting $z$ show sufficient predictive power, and allowing model satisfy Sufficient in Assumption 1. Minimizing Eq. 9 can encourage the model to satisfy the Invariance in Assumption 1.*

## 5 EXPERIMENTS

In this section, we conduct extensive experiments to answer the research questions:

- **RQ1:** How to evaluate the effectiveness of the model in OOD scenarios?
- **RQ2:** How effective is RILOOD in discovering invariant features and improving generalization?

### 5.1 EXPERIMENTAL SETTINGS

**Datasets.** We use six datasets to evaluate our method. Specifically, the Minnesota Solvation Database (MNSolv)(Marenich et al., 2012), QM9Solv(Ward et al., 2021), CompSolv(Moine et al., 2017), ZhangDDI(Zhang et al., 2017b), ChChMiner(Marinka Zitnik et al., 2018) and DeepDDI(Ryu et al., 2018). The detailed statistics and descriptions are given in Appendix B. More experiments are provided in Appendix C.

**Baselines.** We compare our method with the state-of-the-art methods, and adopt 7 baselines: GCN (Kipf & Welling, 2016), CIGIN(Pathak et al., 2020), CGIB(Lee et al., 2023a), CMRL(Lee et al., 2023b), ERM(Vapnik, 2013), GroupDRO (Sagawa et al., 2019) and MixUp(Zhang et al., 2017a).

**Metrics.** We choose widely-used metrics in previous works, the performance of the molecular interaction prediction task is evaluated in terms of RMSE(Pathak et al., 2020) and MAE(Fang et al., 2024). Lower error indicate better prediciton performance. AUROC(Lee et al., 2023b), and Accuracy(Lee et al., 2023b) for DDI prediction.

### 5.2 MAIN RESULTS (RQ1)

#### 5.2.1 REAL-WORLD DATASET

To evaluate the generalization performance of our method, we conducted extensive experiments on three datasets to verify the effectiveness of our proposed method. To explore the possibilities of more environments, i.e. different shifts, we also evaluate performance on different settings: Scaffold, Size, Assay and Solvent. The overall results are summarized in Tab. 1, and we have the following observations:

The results indicate that our method consistently outperforms baseline models, achieving superior performance across all datasets. Conventional methods have limitation as they rely on the core-substructure to generalize, which proves to be an spurious features in molecular pairs relation. The marked improvement in RILOOD can be attributed to its capacity for multi-grained interaction and invariant pattern recognition, which effectively enables the model to adapt to distribution shifts. Further discussions on applying this method to i.d. setting are available in the Appendix C Tab. 3.

Table 1: Performance comparison with baselines on 3 out-of-distribution real-world datasets from MNSolv, CompSolv, QM9Solv in terms of RMSE. Different dataset splits by specific shift. The best and the runner-up results are highlighted in bolded and underlined respectively.

| Method | MNSolv↓ | | CompSolv↓ | | | QM9Solv↓ | |
|---|---|---|---|---|---|---|---|
| | Solvent | Scaffold | Assay | Size | Scaffold | Solvent | Scaffold |
| GCN | $0.8921_{\pm0.024}$ | $1.2752_{\pm0.022}$ | $0.9117_{\pm0.011}$ | $0.7644_{\pm0.024}$ | $0.9598_{\pm0.024}$ | $0.9115_{\pm0.024}$ | $1.0319_{\pm0.024}$ |
| CIGIN | $0.7662_{\pm0.017}$ | $1.3649_{\pm0.021}$ | $0.5299_{\pm0.003}$ | $0.5574_{\pm0.002}$ | $0.6383_{\pm0.005}$ | $0.7503_{\pm0.053}$ | $0.8642_{\pm0.012}$ |
| ERM | $0.7503_{\pm0.026}$ | $\underline{1.3478_{\pm0.013}}$ | $0.5319_{\pm0.011}$ | $0.5360_{\pm0.002}$ | $0.6334_{\pm0.003}$ | $0.7471_{\pm0.053}$ | $\underline{0.7261_{\pm0.005}}$ |
| GroupDRO | $0.7839_{\pm0.003}$ | $1.4322_{\pm0.031}$ | $0.6587_{\pm0.006}$ | $0.5857_{\pm0.013}$ | $0.7459_{\pm0.012}$ | $0.8259_{\pm0.007}$ | $0.8503_{\pm0.021}$ |
| MixUp | $\underline{0.7135_{\pm0.011}}$ | $1.3843_{\pm0.012}$ | $0.5405_{\pm0.022}$ | $0.5772_{\pm0.026}$ | $0.5604_{\pm0.017}$ | $\underline{0.7227_{\pm0.003}}$ | $0.7490_{\pm0.002}$ |
| CGIB | $0.8312_{\pm0.017}$ | $2.2118_{\pm0.024}$ | $0.3916_{\pm0.043}$ | $0.3886_{\pm0.025}$ | $\underline{0.5476_{\pm0.026}}$ | $1.4525_{\pm0.013}$ | $0.7894_{\pm0.006}$ |
| CMRL | $0.8063_{\pm0.012}$ | $2.1524_{\pm0.032}$ | $\underline{0.3865_{\pm0.014}}$ | $\underline{0.3777_{\pm0.023}}$ | $0.6672_{\pm0.013}$ | $1.4425_{\pm0.016}$ | $0.7894_{\pm0.002}$ |
| Ours | $\mathbf{0.6784_{\pm0.007}}$ | $\mathbf{1.0780_{\pm0.013}}$ | $\mathbf{0.3676_{\pm0.017}}$ | $\mathbf{0.3660_{\pm0.022}}$ | $\mathbf{0.5209_{\pm0.014}}$ | $\mathbf{0.7001_{\pm0.001}}$ | $\mathbf{0.6991_{\pm0.003}}$ |

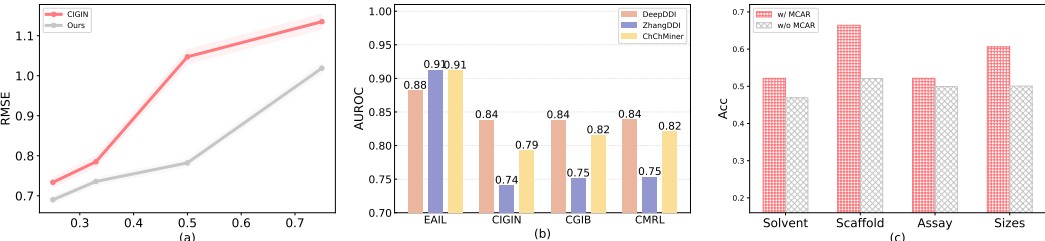

Figure 3: (a) Performance under different spurious correlation levels. We set the strength of spurious correlation as $r = \frac{\text{Number of samples with spurious feature}}{\text{Number of samples}}$, where training set with higher r will have stronger spurious correlations with underlying environments; (b) Results on three DDI datasets with OOD shifts. We conducted comparison experiment with three SOTA methods; (c) Effects of different interaction patterns(w/ MCAR is multi-grained interaction pattern; w/o MCAR is node-level interaction pattern).

### 5.2.2 SYNTHETIC DATASET

We employ different shift strategies tailored to specific datasets, introducing spurious features to create synthetic datasets. We first consider the distribution shift caused by polarity bias w.r.t. eps. The invariant feature is $\widehat{H}_{inv} \in \mathbb{R}$, where $\mathcal{P}(Y|\widehat{H}_{inv})$ is a constant, indicating a stable correlation between Y and $\widehat{H}_{inv}$. Our goal is to learn a model that relies solely on $\widehat{H}_{inv}$. We use eps to control the degree of spurious correlation. The correlation of molecular-pairs and label Y with eps=78 is unstable, counted as E. i.e., $\mathcal{P}(Y|E)$ is unstable, $\mathcal{P}(Y|\widehat{H}_{inv})$ is stable. More detail can be seen in Appedix C Fig. 5. Following(Li et al., 2022; Wu et al., 2022b), the spurious correlation is injected by controlling the variant distribution. Therefore, we manually construct spurious relations of different degrees between $C$ and label Y in the training set. We set r={0.25, 0.33, 0.5, 0.75}. The results are reported in Fig. 3(a). As r grows larger, the performance of all the methods tends to increase since there exists a larger degree of distribution shift. Nevertheless, our proposed method is able to maintain the most relatively stable performance.

### 5.2.3 GENERALIZATION ON GRAPH CLASSIFICATION.

To explore the applicability of our method to other molecular pair data and its potential application in classification tasks, we evaluated its performance on the DDI dataset.The results are reported in Fig. 3(b). RILOOD outperforms previous methods in OOD scenarios as predicted by DDI dataset. This superior performance can be attributed to RILOOD's ability to generalize, effectively transferring knowledge from molecular-pair interactions to molecular with similar interaction patterns and new scaffolds. This transferability ensures that the model remains robust despite distribution shifts.

Table 2: Ablation study on CompSolv-∗ and MNSolv-∗ by RMSE. We show the results of our method that performs best among baselines on all CompSolv-∗ and MNSolv-∗ datasets, for comparison.

| Method | CompSolv↓ | | | | MNSolv↓ | |
|---|---|---|---|---|---|---|
| | Size | Scaffold | Solvent | Assay | Solvent | Scaffold |
| Baseline [B] | $0.5881_{\pm0.010}$ | $0.6383_{\pm0.011}$ | $0.5215_{\pm0.007}$ | $0.5299_{\pm0.023}$ | $0.7662_{\pm0.016}$ | $1.2648_{\pm0.018}$ |
| B + ERM loss [E] | $0.5360_{\pm0.013}$ | $0.5919_{\pm0.012}$ | $0.4864_{\pm0.023}$ | $0.5319_{\pm0.017}$ | $0.7263_{\pm0.026}$ | $1.3478_{\pm0.011}$ |
| B + MCAR [M] | $0.5623_{\pm0.009}$ | $0.5842_{\pm0.003}$ | $0.4914_{\pm0.004}$ | $0.4993_{\pm0.005}$ | $0.7115_{\pm0.003}$ | $1.2191_{\pm0.012}$ |
| M + $\mathcal{L}_{inv}$ [Min] | $0.5598_{\pm0.003}$ | $0.5444_{\pm0.022}$ | $0.5196_{\pm0.003}$ | $0.5483_{\pm0.003}$ | $0.7279_{\pm0.002}$ | $1.2005_{\pm0.003}$ |
| Min + $\mathcal{L}_{MI}$ [MM] | $0.5764_{\pm0.002}$ | $0.5269_{\pm0.013}$ | $0.4980_{\pm0.010}$ | $0.5230_{\pm0.001}$ | $0.6946_{\pm0.008}$ | $1.2360_{\pm0.002}$ |
| Min + $\mathcal{L}_{CVAE}$ [MC] | $0.5482_{\pm0.010}$ | $0.5351_{\pm0.027}$ | $0.4753_{\pm0.023}$ | $0.5188_{\pm0.020}$ | $0.7026_{\pm0.013}$ | $1.1329_{\pm0.011}$ |
| w/o MCAR | $0.6057_{\pm0.009}$ | $0.6641_{\pm0.013}$ | $0.4834_{\pm0.001}$ | $0.5215_{\pm0.013}$ | $0.7285_{\pm0.003}$ | $1.3929_{\pm0.002}$ |
| Ours | $\mathbf{0.3660}_{\pm0.007}$ | $\mathbf{0.5209}_{\pm0.014}$ | $\mathbf{0.4689}_{\pm0.006}$ | $\mathbf{0.3676}_{\pm0.007}$ | $\mathbf{0.6784}_{\pm0.007}$ | $\mathbf{1.0780}_{\pm0.013}$ |

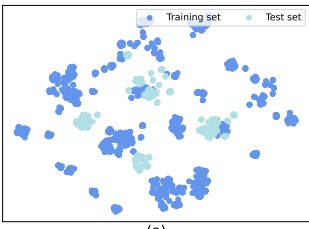 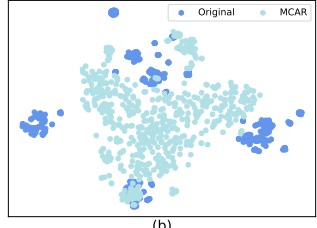 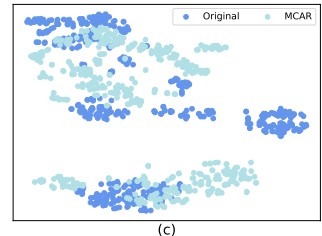

(a)  (b)  (c)

Figure 4: Visualization of the extracted features on training and validation set when the model achieves the best performance on the validation set. (a) The feature distribution of the training set and the test set; (b) Effect of MCAR on solute molecular feature distribution; (c) Effect of MCAR on global feature (solute + solvent) distribution.

## 5.3 IN-DEPTH ANALYSIS (RQ2)

We conduct ablation study by removing the following modules: Multi-granularity Context-Aware Refinement (MCAR) is train by downstream task (M); contrastive loss (Min); condition distribution modeling loss(MC); the removal of MCAR (w/o MCAR); the model is joint-train by Eq. 10 (Ours). The results are presented in Tab. 2. We can observe from the results in Tab. 2 that (1) MCAR plays an important role, especially in invariant learning, which retains not only the original cross-attention but also multilevel attention. (2) Condition modeling plays an important role, but the performance gains of $\mathcal{L}_{CVAE}$ and $L_{MI}$ are much less than for joint training. (3) The removal of MCAR incurs detriment to the overall performance, which illustrates the effectiveness of context interaction.

**Feature Visualization.** To further explore the superiority of our method and understand how multi-granularity context-aware representation remains invariant, we use the t-SNE algorithm to visualize the molecular interactions when the model performs best. For comparison, we also visualized the baseline. As show in Fig. 4, it turns out that (a) The majority of solute molecules in the test set originate from a distinct distribution compared to those in the training set. (b) MCAR has the capacity to enhance the distribution of features, rendering the learning of a more diverse set of features feasible. (c) MCAR is better equipped to capture domain-invariant interaction features, thereby enhancing the model's performance in the unseen domain.

## 6 CONCLUSION

In this paper, we propose an Relational Invariant Learning framework to solve out-of-distribution in molecular relational learning. Three tailored modules are jointly optimized to train the model and learn the representation of invariant molecules in diverse environments. Mixed-enhanced molecular representations are used for variational modeling of diverse environments, further capturing invariant interaction patterns through multi-granularity context-aware refinement strategy. Extensive experiments and theoretical analysis prove the superiority of our method.

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

## A  PROOFS

In this section, we provides detailed proofs in Section 4.

### A.1  PROOF OF EQUATION 12

Considering that the solute features $H_1$ and solvent features $H_2$ in graph-level generated from baseline model are not independent due to the node interaction. Assumed that the solute representations $H_1$ sampled under specific environment and the condition label is c, the mixed representation $\tilde{H}$ of solute and solvent, c, and $G_1$ and $G_c$ are independent, respectively. Then, we have:

$$L(\theta, \phi; \tilde{H}, c) = \mathbb{E}_{q_\phi(z|\tilde{H},c)}[\log p_\theta(\tilde{H}|z,c)] - D_{KL}(q_\phi(z|\tilde{H},c)\|p_\theta(z|\tilde{H})) \tag{11}$$

Here, the aim of learning is to find the best parameter $\theta$ that maximizes the log-likelihood $\log p_\theta(\tilde{H}|c)$. We can derive a tractable variational lower bound known as Evidence Lower BOund (ELBO). Specifically, find the parameters of $p(z|c)$ by minimizing the distance between $p(z|c)$ and $q(z|\tilde{H}, c)$ by KL divergence. Further, we try to maximize the variational lower bound of the log-likelihood $\log p_\theta(\tilde{H}|c)$. Specifically, an auxiliary distribution $q_\phi(z|\tilde{H}, c)$ is introduced to approximate $p_\theta(z|\tilde{H}, c)$, due to the intractability of the true posterior distribution $p_\theta(\tilde{H}|z, c)$.

$$\max_{\theta,\phi} \mathbb{E}_{\mathcal{G}_1 \sim D}\left[\mathbb{E}_{q_\phi(z|\tilde{H},c)}\left[\log p_\theta(\tilde{H} \mid z, c)\right]\right] \text{s.t.} D_{KL}\left(q_\phi(z \mid \tilde{H}, c)\|p_\theta(z|\tilde{H})\right) < \epsilon \tag{12}$$

The threshold $\epsilon$ is primarily to ensure that the learned latent representation $z$ remains faithful to the true underlying distribution of the data. We refer the auxiliary proposal distribution $q_\phi(z|\tilde{H}, c)$ a recognition model and the conditional distribution $p_\theta(\tilde{H}|c, z)$ a generative model. By leveraging approximate posterior inference and reparameterization technique, the prior can effectively capture environmental information from the posterior distribution, thereby facilitating posterior alignment.

$$\log p_\theta(\tilde{H}|c) = -KL(q_\phi(z|\tilde{H},c)\|p_\theta(z|\tilde{H})) + \mathbb{E}_{q_\phi(z|\tilde{H},c)}\left[\log p_\theta(\tilde{H} \mid z, c)\right] \tag{13}$$

where $KL(\cdot\|\cdot)$ is Kullback-Leibler divergence between two distributions. For the regression task, we introduce an uncertainty constraint on the RMSE. Therefore, the reconstructed term is rewritten as:

$$\mathcal{L}_{\text{MCVAE}}(\theta, \phi; \tilde{H}, c) = -KL\left(q_\phi(z^{(l)} \mid \tilde{H}, c)\|p_\theta(z \mid \tilde{H})\right) + \frac{1}{N}\sum_{i=1}^{N}\left[\frac{1}{\sigma(\tilde{H})^2}\|z - \tilde{H}\|^2 + \log \sigma(\tilde{H})^2\right] \tag{14}$$

## B  DATASETS

- **MNSolv** [1] contains 3,037 experimental free energies of solvation or transfer energies of 790 unique solutes and 92 solvents.

- **QM9Solv** [2] contains solvation energies of 130,258 molecules taken from the QM9 dataset computed in five solvents(acetone, ethanol, acetonitrile, dimethyl sulfoxide, and water) via an implicit solvent model. We consider 5,000 experimental free energies of solvation or transfer energies of 1000 unique solutes and 5 solvents.

- **Compsolv** [3] dataset is proposed to show how solvation energies are influenced by hydrogen-bonding association effects. We consider 3,548 combinations of 442 unique solutes and 259 solvents in the dataset following previous work.

---

[1] https://conservancy.umn.edu/bitstream/handle/11299/213300/MNSolDatabase_v2012.zip?sequence=12&isAllowed=y

[2] https://acdc.alcf.anl.gov/mdf/detail/solv_ml_v1.2

[3] https://www.sciencedirect.com/science/article/pii/S0378381210003675

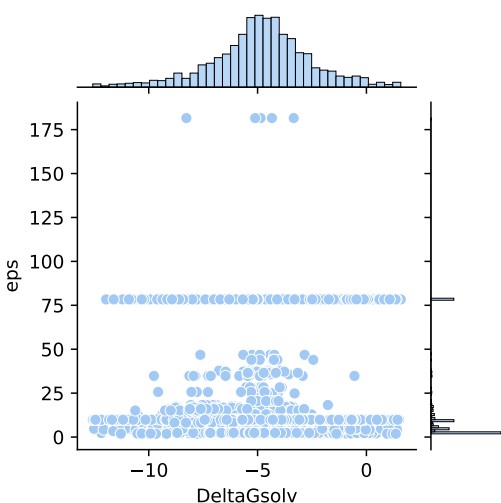

Figure 5: Spurious correlation from dielectric constant $eps$.

- **Abraham**[4] dataset is a collection of data published by the Abraham research group at College London. We consider 6,091 combinations of 1,038 unique solutes and 122 solvents following previous work.

- **Combisolv**[5] contains all the data of MNSol, FreeSolv, CompSolv, and Abraham, resulting in 10,145 combinations of 1,368 solutes and 291 solvents.

## C  ADDITIONAL EXPERIMENTS

### C.1  IMPLEMENTATION AND OPTIMIZATION DETAILS.

The proposed method is implemented on a single NVIDA 3090 GPUs with PyTorch. Following the CIGIN(Pathak et al., 2020), we use the same 3-layer GCN and MPNN as feature extractor for solute molecule and solvent molecule, respectively. More details about backbone can be found in Sec.3.1. During the training, the solute features were incorporated with node interaction features, which is the dot-production similarity between solute node features and solvent node features. Here, we using graph-level solute features and solvent features as input in our method. We select 168 for the dimension ($d_z$) of latent variables. The learning rate was decreased on plateau by a factor of $10^{-3}$ from $10^{-3}$ to $10^{-5}$.

### C.2  GENERALIZATION ANALYSIS.

### C.3  HYPERPARAMETER SENSITIVITY ANALYSIS

We analyze the sensity of the hyperparameters $\alpha$ and $\beta$, which act as the trade-off for loss in Eq.10. In general, the approximate posterior distribution is difficult to approximate the true posterior distribution, resulting in the reconstruction loss being tens to thousands of times that of the supervised loss. In order to balance the rate of decline between individual losses, we performed a hyperparameter sensitivity experiment. The hyperparameter {$\alpha$ is chosen from $10^{-7}, 10^{-6}, 10^{-5}, 10^{-4}, 10^{-3}$}, and in addition, $\beta$ is chosen from {$10^{-8}, 10^{-7}, 10^{-6}, 10^{-5}, 10^{-4}$}. Our experiment is conduct on MNSolvation and CompSolv datasets due the diversity and representativeness of their data. Our results experiences a significant ascend when $\alpha_1$ is large.

---

[4] https://www.sciencedirect.com/science/article/pii/S0378381210003675
[5] https://ars.els-cdn.com/content/image/1-s2.0-S1385894721008925-mmc2.xlsx

Table 3: Performance on molecular interaction prediction task (regression) in terms of RMSE.

| Model | Chromophore | | | MNSolv | CompSolv | Abraham | CombiSolv |
|-------|------------|----------|----------|--------|----------|---------|-----------|
| | **Absorption** | **Emission** | **Lifetime** | | | | |
| GCN | 25.75 | 31.87 | 0.866 | 0.675 | 0.389 | 0.738 | 0.672 |
| GIN | 24.92 | 32.31 | 0.829 | 0.669 | 0.331 | 0.648 | 0.595 |
| CIGIN | 19.32 | 25.09 | 0.804 | 0.607 | 0.363 | 0.472 | 0.451 |
| CGIB | 18.11 | **23.90** | 0.771 | 0.538 | 0.276 | 0.390 | 0.422 |
| CMRL | 17.93 | 24.30 | 0.776 | 0.551 | 0.255 | 0.374 | 0.421 |
| Ours | **17.70** | 25.61 | **0.706** | **0.489** | **0.246** | **0.309** | **0.209** |

Table 4: RMSE result of property prediction task on real-world datasets without/with OOD shifts of domain.

| Dataset | MNSolv | | CompSolv | | QM9Solv | |
|---------|--------|--------|----------|--------|---------|--------|
| **Model** | *w/o OOD* | *w/ OOD* | *w/o OOD* | *w/ OOD* | *w/o OOD* | *w/ OOD* |
| CIGIN | 0.6070 | 0.7662 | 0.3630 | 0.5215 | 0.6932 | 0.7503 |
| CGIB | 0.5380 | 0.8312 | 0.4159 | 0.5678 | 0.3654 | 1.4525 |
| CMRL | 0.5510 | 0.8063 | 0.3363 | 0.8072 | 0.3649 | 1.4425 |
| ERM | 0.5837 | 0.7503 | 0.5290 | 0.6917 | 0.6164 | 0.7471 |
| Mixup | 0.5802 | 0.7135 | 0.4393 | 0.5405 | 0.4268 | 0.7227 |
| **Ours** | **0.4891** | **0.6784** | **0.3497** | **0.5147** | **0.30141** | **0.7001** |

As shown in Fig.6 and Fig.6, we can draw a conclusion that $\alpha$ and $\beta$ plays a role in balancing the trade-off between modeling the environment and invariant learning. In conclusion, different combinations of hyperparameters lead to varying task performance, and we follow the tradition of reporting the best task performance with standard deviations.

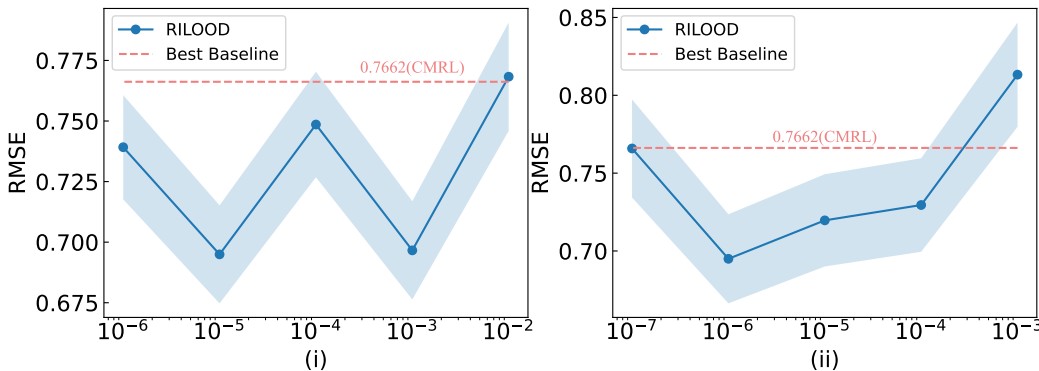

Figure 6: Sensitivity analysis of the hyperparameter (a)$\alpha$ and (b)$\beta$ on $CompSol$ datasets. The solid line shows the average RMSE in the testing stage and the light blue area represents standard deviations. The dashed line represents the average RMSE of the best-performed baseline.

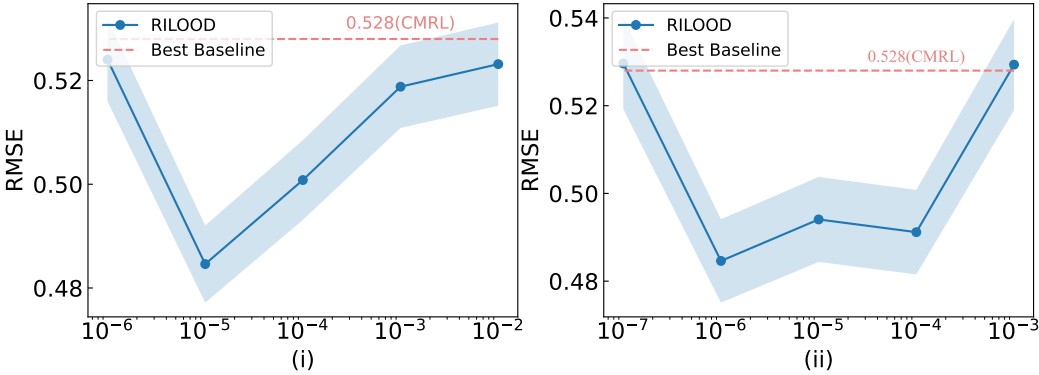

Figure 7: Sensitivity analysis of the hyperparameter (a)$\alpha$ and (b)$\beta$ on $MNSolvation$ datasets. The solid line shows the average RMSE in the testing stage and the light blue area represents standard deviations. The dashed line represents the average RMSE of the best-performed baseline.

Table 5: Hyperparameter specifications.

| | Embedding Dim ($d$) | Batch Size ($K$) | Epochs | Hyperparameter | | |
| --- | --- | --- | --- | --- | --- | --- |
| | | | | lr | $\alpha$ | $\beta$ |
| Absorption | 42 | 32 | 100 | 1e-3 | 1e-3 | 1e-3 |
| Emission | 42 | 256 | 100 | 1e-3 | 1e-3 | 1e-3 |
| Lifetime | 42 | 32 | 100 | 1e-3 | 1e-4 | 1e-3 |
| MNSolv | 42 | 32 | 200 | 1e-3 | 1e-5 | 1e-5 |
| CompSolv | 42 | 256 | 500 | 1e-3 | 1e-6 | 1e-3 |
| Qm9Solv | 42 | 256 | 500 | 1e-3 | 1e-4 | 1e-4 |
| Abraham | 42 | 256 | 500 | 1e-3 | 1e-6 | 1e-6 |
| CombiSolv | 42 | 256 | 500 | 1e-3 | 1e-4 | 1e-3 |
| ZhangDDI | 300 | 512 | 200 | 1e-3 | 1e-3 | 1e-3 |
| ChChMiner | 300 | 512 | 200 | 1e-3 | 1e-4 | 1e-4 |
| DeepDDI | 300 | 512 | 200 | 1e-4 | 1e-4 | 1e-4 |