# OpenReview forum: "Learning Relational Invariance for Out-of-Distribution Molecular Relational Learning"
_ICLR.cc/2025/Conference — ICLR 2025 Conference Withdrawn Submission_

### Official Review · Reviewer_gGAQ · 2024-10-22

**Soundness:** 2
**Presentation:** 2
**Contribution:** 3
**Rating:** 3
**Confidence:** 5

**Summary:**

This paper introduces RILOOD, a framework designed to tackle the out-of-distribution problem within the context of bimolecular interactions. The paper is predicated on the hypothesis that the interactions between molecules are related to certain underlying factors, and it proposes corresponding modules and loss functions to assist the model in learning invariant embeddings for molecule pairs. Experiments across different tasks and datasets are conducted to validate the model's effectiveness.

**Strengths:**

- The topic addressed in this article is highly significant in the context of chemical production practice.
- Extensive experiments across various public benchmarks under out-of-distribution (OOD) settings and ablation studies effectively demonstrate the efficacy of each component.

**Weaknesses:**

- The model demands environment labels for input, which are often difficult to define and sparse in chemical problems. For example, the number of environment labels such as scaffolds increases rapidly with the expansion of dataset size. The method cannot handle the scalability issues brought about by the swift growth of these environment labels.
- The model lacks chemical interpretability, and further clarification regarding the performance gap due to out-of-distribution (OOD) data should be provided. See the questions for details.
- The experiment analysis and writing of the paper should be improved. See questions for details.

**Questions:**

# Major Concerns

## Experiment Settings

- The performance gap between the iid and ood splits is not clearly demonstrated. Table 4 presents the performance differences of various baselines under iid and ood conditions, but these baselines are designed to varying degrees for ood problems. Results for simple backbones, such as the GCN shown in Table 1, should be supplemented.
- Molecular compatibility is highly related to molecular conformation and polarity [1]. This information should not be difficult to learn when given 3D molecular inputs. To strengthen the evaluation, consider demonstrating the performance gap between iid and ood splits using 3D molecular encoders like Uni-mol [3], which can capture conformation and polarity information.

## Experiment Analysis

- Fig. 3 (c) lacks corresponding analysis.
- What is EAIL in Fig. 3 (b)?
- The performance metrics in Table 4 for the CompSolv dataset do not align with those presented in Table 1.
- As shown in Table 2, after adding $\mathcal{L}\_{inv}+\mathcal{L}\_{MI}+\mathcal{L}\_{CVAE}$ and removing MCAR from the baseline, the performance declined. Does this imply that the loss function proposed in this paper leads to a performance degradation,  or that the authors have not accurately reported the experimental results?

##  Chemical Interpretability

- The solubility of molecules has been established to be strongly correlated with properties such as molecular polarity [1]. The authors should supplement the manuscript with relevant experiments or visualizations to demonstrate that the model has indeed captured the strong correlation between polarity and solubility. Otherwise, it remains possible that the model has learned a spurious correlation.
- Additional baselines should be included: [2]. Furthermore, a baseline utilizing DFT calculations or COSMO-RS to compute the Free Energy of Solvation should also be included, given that the dataset is not excessively large so that calculations are feasible.

# Minor Concerns

- This paper introduces a framework that should be compatible with various backbones. However, the experiments in this paper are only conducted on one type of backbone. If time and computational resources permit, we would like the authors to explore the improvements brought by the proposed framework on different backbones.

**If the authors can address all my major concerns, I would be pleased to raise the score.**

# Reference

[1] Anslyn E V. Modern Physical Organic Chemistry[M]. University Science Books, 2006.

[2] Ramani V, Karmakar T. Graph Neural Networks for Predicting Solubility in Diverse Solvents Using MolMerger Incorporating Solute–Solvent Interactions[J]. Journal of Chemical Theory and Computation, 2024, 20(15): 6549-6558.

[3] Zhou G, Gao Z, Ding Q, et al. Uni-mol: A universal 3d molecular representation learning framework[J]. 2023.

---

### Official Review · Reviewer_VN98 · 2024-10-29

**Soundness:** 3
**Presentation:** 2
**Contribution:** 2
**Rating:** 3
**Confidence:** 4

**Summary:**

This paper proposes to learn invariant representation of molecular relational learning.

**Strengths:**

- This paper handles the important problem in the field of molecular science.

- While CMRL discovers causal substructure based on structural causal model, this approach aims to learn invariant representation by introducing environments, which seems to be highly motivated by previous work [1].

[1] Learning Invariant Graph Representations for Out-of-Distribution Generalization, 2022 Neurips

**Weaknesses:**

- In general, I believe writing can be further improved

- Motivation is Not Clear

The explanation of Figure 1 is not clear.
The existing work (CMRL) is also based on a causal framework to remove spurious correlated substructures. Why is the solute feature being referred to as a spurious correlation?

Even if, as the authors claim, CMRL injects noise into the representation space, this paper also performs reconstruction in the representation space without imposing specific constraints or conditions on the functional group. Therefore, it is not significantly different from CMRL.

- Assumption Section

It is stated that the (G_1, G_2) pair is related to the surrounding environment of K, but what examples are there of the surrounding environment? In the modeling section (lines 251-253), it is stated that the solvent is the environment, and it seems that the solvent acts as the environment in the modeling section as well.

- Experiment Section

A more detailed explanation of the experimental setup is needed. For example, Solvent, Scaffold, Assay, and Size seem to be methods of data splitting, but how were these obtained?
The settings vary slightly for each experiment, and it needs to be clarified why that is. For example, in Table 1, CompSolv experiments on Assay, Size, and Scaffold splits, while in Table 2, experiments are conducted on Size, Scaffold, Solvent, and Assay.

- No codes are provided, raising question on reproducibility of this work.

**Questions:**

- Equation 3: What are c1 and c2? I understand H1 as the solute and H2 as the solvent, but is c1 then the solute? If not, it would be good to explain how c1 and c2 were sampled.

- Section 4.2: What is the dimension of z?

- Line 303: Clarifying the dimension of E would be helpful. H2 has an N^2 X 2D dimension, but the process of multiplying these with the transformation matrices (W) is not well understood.

**Details Of Ethics Concerns:**

While reading the appendix, I noticed that the Abraham and Combisolv datasets were not used in the main task experiments. Even though I later confirmed it was in the appendix, I found that the description from the CGIB paper was used verbatim, which could lead to issues related to plagiarism.

---

### Official Review · Reviewer_dCgX · 2024-11-02

**Soundness:** 2
**Presentation:** 2
**Contribution:** 2
**Rating:** 5
**Confidence:** 4

**Summary:**

This paper introduces RILOOD, a framework that enhances the generalization of molecular property predictions to new environments. Specifically, it leverages multi-granularity interactions and mixup-based conditional modeling to capture invariant patterns across molecular environments.

**Strengths:**

- RILOOD is a pioneering approach specifically designed for out-of-distribution (OOD) generalization in Molecular Relational Learning (MRL).
- The approach sounds plausible.

**Weaknesses:**

- Personally, the method is not very innovative.
- I believe the baselines for comparison are not strong enough, particularly when it comes to methods for drug-drug interaction (DDI).
- The paper also does not utilize datasets that are commonly employed in DDI tasks.

**Questions:**

I hope the authors can provide additional experimental results.

**Minor concerns:**
- It seems that the bottom part of Figure 2 has been inadvertently cropped.
- I find Figure 1 to be somewhat confusing.
- Around line 695-696, there is a misplaced parenthesis in the expression for the hyperparameter $\alpha$.

---

### Official Review · Reviewer_HEy7 · 2024-11-04

**Soundness:** 2
**Presentation:** 1
**Contribution:** 2
**Rating:** 3
**Confidence:** 4

**Summary:**

The authors argue that previous approaches to predicting molecular relational properties in out-of-distribution (OOD) cases fall short, as they fail to effectively distinguish invariant causal components from spurious correlations.
To address this issue, the authors propose a Relational Invariant Learning framework called RILOOD, which learns environment-conditioned representations using a conditional variational autoencoder (CVAE) and captures molecular-level and feature-level contexts through coarse-grained and fine-grained extractors, respectively.
Through experiments conducted in an OOD setting, the authors demonstrate the effectiveness of the proposed model.

**Strengths:**

- The authors provide a formal definition of the out-of-distribution (OOD) case in molecular relational learning
- The proposed model shows robust performance in the OOD scenarios.

**Weaknesses:**

- The writing of this paper is not well organized and lacks important details. Specifically, the explanation of the toy example, which is crucial for conveying the paper's motivation, is insufficiently detailed. I have included questions about ambiguous points in the 'Questions' section.

- The authors conducted out-of-distribution (OOD) experiments by splitting the train and test sets based on scaffold, size, assay, and solvent. However, the details of how this splitting was performed are not provided.

- In the MCVAE module, the author proposes categorizing the solvent into K categories; however, the details of what these categories entail are not defined. Are these categories the same as the environmental factors mentioned in the experiments section (i.e., scaffold, size, assay, etc.)? If they are the same, I am curious whether the performance would vary significantly if they are not.

- While achieving robust performance in out-of-distribution (OOD) scenarios is a significant contribution of this paper, the details of how the authors constructed the synthetic dataset are not well defined, particularly the origin of the data. Additionally, the comparison with CIGIN is inadequate; please include all baseline methods employed in these experiments.

**Questions:**

- The explanation of the toy example (Figure 1) lacks sufficient detail for clear understanding. The authors should expand on this section to provide more clarity.

- In line 172, the author states that $H_1 \in \mathbb{R}^{N_1 \times D}$, which implicitly suggests that $H_2 \in \mathbb{R}^{N_2 \times D}$ as well. However, in Equation (3), it is unclear how $H_1$ and $H_2$ can be added given their differing dimensions.

- In the last sentence of the first paragraph of Section 4.2, the authors state, "Consequently, we utilize auxiliary information as a condition, such as solvent, to model the distribution of molecules across domains." However, they do not clarify what this auxiliary information entails.

- In the last paragraph of page 6 (line 319), $\hat{H}_{inv}$ is mentioned without prior definition; it appears to correspond to $H_c$ in equation 7.

- In Table 2, why is the QM9Solv dataset excluded?

---

### Note · Authors · 2024-11-22

I have read and agree with the venue's withdrawal policy on behalf of myself and my co-authors.